# The Anti-CGRP Antibody Fremanezumab Lowers CGRP Release from Rat Dura Mater and Meningeal Blood Flow

**DOI:** 10.3390/cells11111768

**Published:** 2022-05-28

**Authors:** Mária Dux, Birgit Vogler, Annette Kuhn, Kimberly D. Mackenzie, Jennifer Stratton, Karl Messlinger

**Affiliations:** 1Department of Physiology, University of Szeged, H-6720 Szeged, Hungary; dux.maria@med.u-szeged.hu; 2Institute of Physiology and Pathophysiology, Friedrich-Alexander-University, D-91054 Erlangen-Nürnberg, Germany; birgit.vogler@fau.de (B.V.); annette.kuhn@fau.de (A.K.); 3Teva Pharmaceuticals, Redwood City, CA 94063, USA; kimberly.mackenzie01@tevapharm.com (K.D.M.); jennifer.stratton@tevapharm.com (J.S.)

**Keywords:** fremanezumab, monoclonal antibody, calcitonin gene-related peptide, glycerol trinitrate, CGRP release, meningeal blood flow, rat, migraine pain

## Abstract

Monoclonal antibodies directed against the neuropeptide calcitonin gene-related peptide (CGRP) belong to a new generation of therapeutics that are effective in the prevention of migraine. CGRP, a potent vasodilator, is strongly implicated in the pathophysiology of migraine, but its role remains to be fully elucidated. The hemisected rat head preparation and laser Doppler flowmetry were used to examine the effects on CGRP release from the dura mater and meningeal blood flow of the subcutaneously injected anti-CGRP monoclonal antibody fremanezumab at 30 mg/kg, when compared to an isotype control antibody. Some rats were administered glycerol trinitrate (GTN) intraperitoneally to produce a migraine-like sensitized state. When compared to the control antibody, the fremanezumab injection was followed by reduced basal and capsaicin-evoked CGRP release from day 3 up to 30 days. The difference was enhanced after 4 h of GTN application. The samples from the female rats showed a higher CGRP release compared to that of the males. The increases in meningeal blood flow induced by acrolein (100 µM) and capsaicin (100 nM) were reduced 13–20 days after the fremanezumab injection, and the direct vasoconstrictor effect of high capsaicin (10 µM) was intensified. In conclusion, fremanezumab lowers the CGRP release and lasts up to four weeks, thereby lowering the CGRP-dependent meningeal blood flow. The antibody may not only prevent the released CGRP from binding but may also influence the CGRP release stimulated by noxious agents relevant for the generation of migraine pain.

## 1. Introduction

The sensory neuropeptide calcitonin gene-related peptide (CGRP) is considered to be crucially involved in the generation and aggravation of migraine and trigemino-autonomic headaches [1,2,3]. Increased levels of CGRP have been found during migraine and cluster headache attacks in the venous outflow from the head and in peripheral blood [4,5,6,7]. Conversely, the infusion of CGRP caused delayed migraine-like headaches in migraineurs and cluster-like attacks in patients suffering from cluster headaches, respectively [8,9]. Targeting CGRP signalling has long been proven to be effective in the treatment of migraine. Triptans reduce CGRP release, and new small-molecule CGRP receptor antagonists have beneficial effects in migraine therapy [10,11,12,13]. Consistently, in preclinical models of migraine, CGRP-targeting antibodies or CGRP receptor antagonists have proven to be effective in reducing elevated trigeminal activity [14,15,16,17].

For more than five years now, monoclonal antibodies targeting CGRP or its receptor have shown their efficacy in preventing chronic, frequent, and episodic migraine [18,19,20]. Thus, it is quite evident that the attenuation of the CGRP signalling system is effective in reducing the occurrence and severity of migraine attacks; however, neither the sites nor the mechanisms of migraine inhibition or prevention are sufficiently elucidated. There are good reasons to assume that the main effects are peripheral as monoclonal antibodies cannot readily cross the blood-brain barrier (BBB) [21,22]. Much is still to be discovered regarding the mechanisms of CGRP action and the prevention of its signalling on trigeminal afferents, which causes sensitization in both migraine-related preclinical studies [14,23,24,25,26,27] and in patients suffering from primary headaches, triggering their specific types of headaches [9,28,29,30,31].

CGRP is present in a major proportion of spinal and trigeminal nociceptive afferents [11,32,33,34,35] and is released from activated peripheral and central terminals and possibly also from cell bodies within the sensory ganglia [35,36,37]. Most of the CGRP-containing trigeminal neurons are chemosensitive in nature, expressing the nociceptive cation channels transient receptor potential vanilloid (TRPV1) and the transient receptor potential ankyrin 1 (TRPA1) receptors. The basal and stimulated release of CGRP and substance P from rodent meningeal tissues has been used as an approved method that reflects the state of activity of trigeminal afferents [37,38,39,40,41,42,43,44]. In these studies, inflammatory mediators, such as high concentrations of potassium chloride (depolarizing agent), capsaicin (TRPV1 agonist), or acrolein (TRPA1 agonist), have been applied for stimulation. In clinical experiments, the infusion of CGRP or the nitrovasodilator glycerol trinitrate (GTN, nitroglycerin) induced immediate headaches and increased CGRP plasma levels [45,46]. In addition, these substances triggered delayed migraine-like headaches nearly exclusively in migraine patients [28,47]. Therefore, GTN infusion or injection has frequently been used to model meningeal nociception and migraine-like states in animal experiments [48,49,50]. Under experimental conditions, the injection of GTN resulted in an increased proportion of the trigeminal neurons that are immunoreactive for CGRP and neuronal NO synthase [51].

As a read-out of the functional condition of the trigeminal nociceptors and released CGRP, the vasodilatation of meningeal arteries and the changes in meningeal blood flow have been recorded in rodent in vivo models [52,53,54]. CGRP is the most effective vasodilator substance acting on arterial meningeal vessels [55] and is therefore mainly responsible for increased meningeal blood flow when the trigeminal afferents are activated to release their neuropeptides. Under experimental conditions, trigeminal afferents have been stimulated electrically and chemically, using substances such as acrolein and capsaicin [43,56,57,58,59,60,61]. These substances are well-known agonists at the TRPA1 and TRPV1 receptor channels, respectively, releasing neuropeptides upon calcium influx. From previous experiments, we determined that the meningeal blood flow response to capsaicin depends on the capsaicin concentration; at nanomolar concentrations, capsaicin causes arterial dilatation and increased blood flow, but at micromolar concentrations, it decreases meningeal perfusion [62]. Thus, capsaicin exerts a dual function on meningeal blood vessels; by releasing CGRP from trigeminal afferents, it increases meningeal blood flow, while capsaicin acting on vascular TRPV1 receptors constricts smooth muscle cells, reducing meningeal blood flow.

Further exploration of the mechanisms of the CGRP effect in trigeminal pathophysiology guided the rationale of the present preclinical study using fremanezumab [63,64,65]. We do not know exactly where the anti-CGRP antibodies exert their effect within the trigeminal tissues but assume that the cranial dura mater is an important site of action [21,66,67]. Systemically applied anti-CGRP antibodies have a long-lasting migraine prevention effect [68,69]. We therefore speculate that fremanezumab not only neutralizes the CGRP released into the tissues but may also reduce the amount of CGRP released in the trigeminovascular system upon stimulation. We used in vivo and ex vivo rat dura mater preparations to measure the CGRP release from the trigeminal afferents and the consequent changes in meningeal perfusion following the pre-administration of fremanezumab, when compared to an isotype control antibody.

## 2. Materials and Methods

The animal housing and all the experiments were carried out according to the German guidelines and regulations of the care and treatment of laboratory animals and the European Communities Council Directive of 24 November 1986 (86/609/EEC), amended 22 September 2010 (2010/63/EU). The experimental protocols were reviewed by an ethics committee and approved by the District Government of Middle Franconia (54-2532.1-21/12).

### 2.1. Animals

Adult Wistar rats of both sexes (body weight of females: 230–370 g; males: 230–450 g), bred and housed in the animal facility of the Institute of Physiology and Pathophysiology of the FAU Erlangen-Nürnberg, were used. They were kept in a 12 h light/dark cycle in standard cages in groups of 3–4 and fed with standard food pellets and water ad libitum. The animals were matched and distributed according to their gender and weight, as equally as possible, for the different experiments. We also matched the two antibodies used and the observation time after administration over the whole experimental period as far as possible. The oestrus state of the females was not assessed.

### 2.2. Administration of Antibodies

The rats were anaesthetized around 9 a.m. in a plastic box with a concentration of isoflurane increasing up to 4% (Forene, Abott, Wiesbaden, Germany), applied with an evaporator (Forane Vapor 19.3, Dräger AG, Lübeck, Germany). The animals were weighed, and the neck region was shaved and disinfected with 70% ethanol. Then, 30 mg/kg anti-CGRP antibody fremanezumab or isotype control antibody (Teva Pharmaceuticals, Redwood City, CA, USA) diluted in saline (10 mg/mL) was subcutaneously injected in an even distribution 2 cm left and right from the midline and 5 cm from the caudal of the occiput, using a syringe with a 27-gauge needle. A human IgG2 antibody-targeting keyhole limpet hemocyanin (KLH) was used as the isotype control antibody to assess the specific effect of the targeting CGRP. The examiners were blinded as to the identity of the antibodies. The rats were marked at their tail for identification and placed back in their cage, where they recovered from the anaesthesia usually within 2–3 min. The animals were inspected two times on every following day with regard to any unusual behaviour.

### 2.3. Preparation for CGRP Release Measurements

On day 1, 3, or 10 after the antibody injection at around 9 a.m., the rats were again shortly anaesthetized by isoflurane to receive an intraperitoneal (i.p.) injection of 5 mg/kg glycerol trinitrate (GTN, 1 mg/mL in saline) or the same volume of saline as a vehicle, equal to the GTN solution, using a 23 G needle. Four hours later, the rats were deeply anaesthetized and killed in an atmosphere of an increasing concentration of CO_2_. The head was separated, skinned, and divided in the midline, and the two skull halves with adhering dura mater were prepared for the measurement of the CGRP release according to a standard protocol [39]. The skull halves were washed for 30 min with synthetic interstitial fluid (SIF) and mounted in a water bath above warm water (37 °C), holding the temperature constant. The SIF was composed of (in mM): 107.8 NaCl, 3.5 KCl, 0.69 MgSO_4_ 7 H_2_O, 26.2 NaHCO_3_, 1.67 NaH_2_PO_4_ 2 H_2_O, 9.64 Na-gluconate, 5.55 glucose, 7.6 sucrose, and 1.53 CaCl_2_ · 2 H_2_O buffered to pH 7.4 with carbogen gas (95% O_2_, 5% CO_2_). The skull halves were filled twice with 300 µL of SIF, followed by a solution of 500 nM capsaicin (dissolved in saline with 1% ethanol and further diluted with SIF) and another SIF; all the applications were at intervals of 5 min. The chosen capsaicin concentration exerts a robust CGRP release [70]. At the end of each interval, the fluid was carefully collected using a pipette without touching the tissue.

In some experiments, the long-term effect of the fremanezumab treatment on the stimulated CGRP release was measured. In these experiments, the meningeal afferents were stimulated by capsaicin (500 nM) application 28–30 days after antibody treatment and 4 h after GTN injection.

### 2.4. Analysis of Released CGRP Concentration

From the collected fluid samples, 100 µL was separated; immediately, 25 µL of enzyme-immunoassay (EIA) buffer containing peptidase inhibitors (Bertin Pharma/SPIbio, Montigny le Bretonneux, France) was added. The samples were deep-frozen and stored at −20 °C until their analysis, together with the samples of further experiments. After thawing, the samples were processed using an EIA kit for CGRP according to the instructions of the manufacturer (Bertin Pharma/SPIbio, Montigny le Bretonneux, France). The EIA is based on a double-antibody sandwich technique, with monoclonal capture and tracer antibodies binding the CGRP molecule; the tracer antibody is conjugated with acetylcholine esterase converting Ellman’s reagent to a yellow substance, the absorbance of which is measured by a photo-spectrometer (Opsys MR, Dynex Technologies, Denkendorf, Germany). The assay has 100% reactivity for rat CGRP but <0.01% cross-reactivity with other proteins of the calcitonin family and detects both *α*- and *β*-CGRP with the same sensitivity. The lower limit of detection is 2 pg/mL according to the manufacturer’s information. The CGRP concentrations in the original samples were calculated in pg/mL, considering the added volume of EIA buffer.

### 2.5. Preparation for Meningeal Blood Flow Recordings

Between day 13 and day 20 after the antibody injection, the rats were again anaesthetized by 4% isoflurane, followed by an application of 2% isoflurane through a tight mask. Atropine sulfate (B. Braun Melsungen AG, Melsungen, Germany, 0.5 mg/mL 1:10 with sodium chloride 0.9%) was injected subcutaneously to prevent salivation. The animals were tracheotomized in order to be artificially ventilated with a mixture of oxygen-enriched room air and 2% isoflurane. The depth of anaesthesia was routinely assessed and held at a level in which noxious stimuli (pinching of earlobes) failed to elicit motor reflexes or changes in systemic arterial pressure. The body temperature of the animals was recorded by a thermoprobe inserted into the rectum and was kept at 37–37.5 °C with a feedback-controlled heating pad. Systemic blood pressure was recorded with a pressure transducer connected to the catheter inserted into the right femoral artery. The expiratory CO_2_ was continuously monitored (Artema MM 200, Karl Heyer, Bad Ems, Germany) and maintained at 3–3.5%. The head of the animal was fixed in a stereotaxic frame and held by ear bars and a snout clamp. The eyes were covered with dexpanthenol ointment (Bepanthen^®^, Bayer Vital GmbH, Leverkusen, Germany) to prevent dehydration of the cornea.

### 2.6. Meningeal Blood Flow Recordings

A median incision was made along the midline of the scalp, the periosteum was moved aside, and a cranial window of about 8 × 6 mm was drilled into the parietal bone under saline rinsing to expose the cranial dura mater. The exposed dura mater in the parietal cranial window was covered with SIF. Two needle type probes of a laser Doppler flowmeter (DRT4, Moor Instruments, Axminster, UK) were positioned over the branches of the middle meningeal artery supplying the dura mater. Blood flow was recorded at a sampling rate of 10 Hz and expressed in arbitrary perfusion units (AU), which apply to the output voltage (mV) of the flowmeter. The systemic blood pressure was recorded simultaneously. The data were stored and processed with the MoorSoft program for Windows. For chemical stimulation, the SIF was replaced by 40 µL of solutions, which were washed off after 5 min, 3 times with SIF, and replaced by the next solution 10 min after the last SIF application. The solutions were applied in a fixed order: SIF, acrolein 100 µM, SIF, capsaicin 100 nM, SIF, and capsaicin 10 µM. The basal blood flow was the mean flow value measured during a 3 min period prior to the stimulation of the dura mater. The blood flow values during stimulation were assessed as mean values, measured during 5 consecutive 1 min periods and during the whole 5 min application period, and were compared to the respective basal flow measured prior to stimulation.

### 2.7. Data Processing and Statistics

Statistical analysis was performed on non-normalized values using Statistica software (StatSoft, Release 7, Tulsa, OK, USA). Following verification of the normal distribution of data, the Student’s *t*-test and analysis of variance (repeated measures or factorial ANOVA) were used and extended by Tukey’s honest significant difference (HSD) test or Fisher’s least square difference (LSD), as specified in the results. The level of significance was set at *p* < 0.05. The data are displayed as mean ± SEM (standard error of the mean).

## 3. Results

### 3.1. Tolerability of Treatments

The antibodies were injected subcutaneously into 68 rats (33 females and 35 males) and allocated equally to the fremanezumab and the isotype control antibody groups. The injection of either antibody did not cause any unusual behaviour during the following days. In 34 animals, GTN was i.p. injected on the day of the experiment. The injections were not followed by any unusual behaviour or licking of the injection sites. No signs of irritation or inflammation at the injection sites were observed. After GTN injection, the rats were not observed to show any unusual behaviour or decrease in their motor activities. None of the animals died during the waiting time following the injection of antibodies.

### 3.2. Body Weight

The rats in all the groups gained body weight during the experiment; the males with the higher body weight (on average 325.6 g) gained 13.8 ± 5.0 g after 3 days and 53.3 ± 5.6 g after 10 days; the females (269.4 g) gained 7.5 ± 1.6 g after 3 days and 17.1 ± 3.5 g after 10 days. Using factorial ANOVA, there was a significant difference between the sexes (F_1,52_ = 11.72, *p* < 0.005) and between the waiting days after antibody treatment (F_2,52_ = 30.14, *p* < 0.0001), as expected due to general body growth, but not between the fremanezumab- and the isotype control antibody-treated groups (F_1,52_ = 1.27, *p* = 0.266).

### 3.3. CGRP Release from the Dura Mater

#### 3.3.1. Exclusion of Antibody-Assay Interactions

To test whether the antibodies interacted directly with the EIA kit, 30 µL of fremanezumab (10 mg/mL in saline), or the control antibody at the same dose, was diluted with 70 µL SIF and 25 µL EIA buffer and processed. The apparent CGRP concentration of the fremanezumab solution was 6.2 pg/mL, that of control antibody was 8.4 pg/mL. The apparent CGRP concentration of SIF alone (*n* = 4) was 8.3 ± 0.71 pg/mL, i.e., values below 8 pg/mL reflect a virtually zero CGRP concentration according to the manufacturer. Thus, neither the fremanezumab nor the isotype control antibody interacted directly with the assay to influence the virtual CGRP concentration in the solution.

#### 3.3.2. Exclusion of Side Difference

Equal numbers of males and females (total *n* = 48) were used for the release experiments. First, we tested whether there was a systematic difference between the two skull halves of the animals, which were both used for examining the CGRP release from the dura mater. Therefore, we applied ANOVA with repeated measurements to compare the sequential release values of all the experiments and set the side of the head (left-right) as the independent factor (Figure 1A). As expected, the sequential CGRP values varied significantly (F_3,282_ = 682.61, *p* < 0.0001), with a significant increase following the capsaicin application (Tukey post hoc test, *p* < 0.0001; Figure 1A, ***), but there was no difference between the two basal CGRP levels measured at 5 and 10 min (Tukey, *p* = 1.00). There was also no difference between the two skull halves of the same animals (F_1,94_ = 0.19, *p* = 0.597). Therefore, the factor side was eliminated from further statistical evaluations, and the measurements of both sides were independently used for the further analysis of the other factors, i.e., the numbers used for statistics are the numbers of the skull halves.

#### 3.3.3. Impact of Antibodies on CGRP Release

We analysed whether the type of the antibody pre-treatment had an impact on the CGRP release in the course of the experiment, irrespective of the other factor treatments, the sex, and the waiting time (Figure 1B). The basal CGRP release (mean of the measurements at 5 and 10 min) was 19.2 ± 0.9 pg/mL in the animals injected with the control antibody (*n* = 24) and 15.6 ± 0.7 pg/mL in the animals injected with fremanezumab (*n* = 24). The application of capsaicin (500 nM) was followed by a 10-fold increase, approximately, in CGRP release in all groups. The mean (± SEM) of all the stimulated release values was 190.9 ± 8.2 pg/mL in the animals treated with the control antibody and 137.5 ± 5.8 pg/mL in the animals treated with fremanezumab; the difference is clearly significant (repeated measures ANOVA and Tukey test, F_1,94_ = 32.14, *p* < 0.001; Figure 1B, ##). After removing the capsaicin solution, the CGRP release values fell to 40.2 ± 1.1 pg/mL in the animals treated with the control antibody and 31.0 ± 1.1 pg/mL in the animals treated with fremanezumab.

#### 3.3.4. Impact of Treatment, Sex, and Waiting Time on Basal CGRP Release

In addition to the two antibody types, each of the other factors (GNT/vehicle injection, sex, and day after antibody injection) seemed to influence the basal CGRP release (Figure 2). The factorial ANOVA and the Tukey post hoc tests were used to test the differences in basal CGRP release (mean of values at 5 and 10 min) specifically caused by these factors interacting with the factor antibody. In the animals pre-treated with the isotype control antibody (*n* = 24), the basal CGRP release was lower after injection with GTN compared to the vehicle injection (F_1,72_ = 8.07, *p* < 0.01; Figure 2A, *), which was aggravated in the animals (*n* = 24) pre-treated with fremanezumab (*p* < 0.001; Figure 2A, **). In the control animals, a lower basal CGRP release was also found in the males compared to the females (F_1,72_ = 6.68, *p* < 0.001; Figure 2B, **). In addition, a lowered basal CGRP release was found after waiting 3 and 10 days compared to 1 day (F_2,72_ = 0.35, *p* < 0.001; Figure 2C, **). Following the fremanezumab pre-treatment, the basal CGRP release appeared to be lower in most groups compared to the rats treated with the isotype control antibody (F_1,72_ = 31.55, *p* < 0.001; Figure 2A–C). The difference was statistically significant after GTN injection (*p* < 0.001; Figure 2A, ##), in females (*p* < 0.001; Figure 2B, ##), and after waiting times of 3 and 10 days, *p* < 0.05; Figure 2C, #).

#### 3.3.5. Impact of Treatment, Sex, and Waiting Time on Stimulated CGRP Release

As for the basal release, each of the other factors (GTN injection, sex, and days) may have had an influence on the stimulated CGRP release (Figure 3). Factorial ANOVA extended by the Tukey post hoc test was used to test differences in the capsaicin-evoked CGRP release (at 15 min) specifically caused by these factors interacting with the factor antibody. After the GTN injection, the capsaicin-induced increase in CGRP release was higher compared to the vehicle-injected samples only in the group of animals pre-treated with the isotype control antibody (F_1,72_ = 7.70, *p* < 0.01; Figure 3A, *). Likewise, the capsaicin-induced CGRP release was higher in the female rats, but only in these control antibody-treated animals (F_1,72_ = 9.58, *p* < 0.001; Figure 3B, **). The stimulated CGRP release was lower in all the groups of animals that had received fremanezumab compared with the control antibody (Figure 3A–C). The lowering effect of fremanezumab was more significant after the GTN injection (*p* < 0.001) than after the vehicle (*p* < 0.05) (Figure 3A, ## and #, respectively) and in the females (*p* < 0.001) than in the males (*p* < 0.05) (Figure 2B, ## and #, respectively). Regarding the waiting time, the CGRP release fell from day 1 to day 3 only in the fremanezumab-treated group (*p* < 0.05; Figure 1C, *) but was stable in the isotype control antibody-treated group. Furthermore, the lowering CGRP release following fremanezumab compared to the control antibody was already significantly reduced at day 1 (*p* < 0.05), with higher significance at days 3 and 10 (*p* < 0.001) (Figure 3C, # and ##, respectively).

Thus, fremanezumab lowered the capsaicin-evoked CGRP release, particularly after the GTN treatment and in female animals after a waiting time of 3 days.

#### 3.3.6. Additional Experiments with Longer Waiting Time

In addition to the main release experiments, four female and six male animals were used to examine the CGRP release from the skull halves 28–30 days after the antibody injection and 4 h after the GTN treatment. The capsaicin-evoked CGRP release in females was 177.1 ± 17.2 pg/mL after pretreatment with the isotype control antibody and 154.9 ± 5.0 pg/mL after fremanezumab; in males, the respective values were 318.1 ± 30.8 pg/mL and 185.9 ± 16.5 pg/mL. Factorial ANOVA extended by the Tukey post hoc test, using the factors of antibody and sex, showed significant differences both for the antibody (ANOVA, F_1,16_ = 11.28, *p* < 0.01) and the sex (F_1,16_ = 13.99; *p* < 0.01). Thus, the fremanezumab injection reduced the stimulated CGRP release for at least 4 weeks.

### 3.4. Meningeal Blood Flow

#### 3.4.1. Basal Blood Flow

The basal blood flow depends largely on the sites where the probes are positioned, i.e., it is generally higher when larger arteries are measured. Because the maximal flow value is the 1000 arbitrary units (AU) that can be displayed by the flowmeter, we aimed to set the probes onto sites with a flow value that warranted a wide scope of flow changes, which is usually the case at one of the main branches of the middle meningeal artery. Thus, most of the values were between 250 and 500 AU. For the flow measurements, 10 animals (5 females and 5 males) were used. In each animal, the blood flow was measured with two flow probes, which were positioned on different arteries, yielding two independent measurements; the number of measurements related to the application of the substances is seen in Figure 4 and Figure 6. The basal blood flow was compared with factorial ANOVA, including the factors of sex, antibody, and stimulation (acrolein 100 µM, capsaicin 100 nM and 10 µM) extended by the Tukey HSD post hoc test. Regarding the basal flow, irrespective of the type of antibody treatment, there was a significant difference between the females (mean ± SEM: 301.8 ± 33.4 AU) and the males (403.2 ± 22.5 AU) (F_1,58_ = 8.87, *p* < 0.01). There was also a difference between the animals treated with the control antibody (mean ± SEM: 403.4 ± 26.5 AU) and fremanezumab (318.5 ± 28.5 AU) (F_1,58_ = 5.55, *p* < 0.05). The interaction between the sexes and the antibodies was also significant (F_1,58_ = 26.06, *p* < 0.001), and the Tukey post hoc test showed that the difference between the antibodies was solely based on a difference within the females (*p* < 0.001) but not the males (*p* = 0.337) (Figure 4A). Finally, there was no difference in the baseline flow values in the same animal before the serial application of acrolein and capsaicin at the two concentrations (F_1,58_ = 1.28, *p* = 0.28) (Figure 4B). Thus, the basal flow was lower in the female animals after treatment with fremanezumab, probably due to the lower unstimulated basal CGRP release.

#### 3.4.2. Stimulated Blood Flow

##### Stimulation with Acrolein

After application of the TRPA1 receptor agonist acrolein (100 µM), the meningeal blood flow increased slightly in the rats treated with the control antibody but did not significantly change in the animals treated with fremanezumab (Figure 5A and Figure 6A left). Two-way repeated measures ANOVA of single-minute values (factor time) and the factor antibody showed a significant change over time (F_5,110_ = 4.38, *p* < 0.005) and a significant difference between the baseline and the values of minutes 3–5 in the control antibody experiments (LSD post hoc test, *p* < 0.05). In the fremanezumab-treated animals, the meningeal blood flow decreased transiently within the first minute (*p* < 0.05). The mean flow during the 5 min acrolein stimulation was 102.3% of the baseline in the control antibody experiments and 98.3% in the fremanezumab experiments (Figure 6A right). Thus, acrolein caused a moderate increase in the meningeal blood flow with a delay of 2–3 min in the rats treated with the antibody isotype but not in the animals treated with fremanezumab.

##### Stimulation with Low-Dose Capsaicin

After application of the TRPV1 agonist capsaicin (100 nM), the blood flow tended to decrease transiently and then increased in the rats treated with the control antibody but showed no significant change in the rats treated with fremanezumab (Figure 5B and Figure 6B left). Two-way repeated measures ANOVA showed a significant change over time (F_5,110_ = 8.75, *p* < 0.0001) and a significant difference between the baseline and the values of minutes 4 and 5 in the isotype control antibody experiments (LSD post hoc test, *p* < 0.05); there was no difference in any minute of the fremanezumab experiments. The mean blood flow during the 5 min capsaicin stimulation was 103.9% of the baseline in the control antibody experiments and 98.6% in the fremanezumab experiments (Figure 6B right). Thus, capsaicin at the low dose of 100 nM caused an increase in meningeal blood flow after 3–4 min of application in the rats treated with the isotype control antibody but not in the animals treated with fremanezumab.

##### Stimulation with High-Dose Capsaicin

After the application of capsaicin at 10 µM, which, in addition to the CGRP-releasing effect, vigorously stimulates the TRPV1 receptors of the smooth muscle cells of the meningeal blood vessels, the flow decreased transiently in the rats treated with the control antibody and permanently in the animals treated with fremanezumab (Figure 5C and Figure 6C left). Two-way repeated measures ANOVA showed a significant change over time (F_5,110_ = 17.07, *p* < 0.0001) and a significant difference in the interaction of the factors of time and antibody (F_5,110_ = 5.59, *p* < 0.0005). The post hoc LSD test indicated that in the control antibody experiments, the values of minutes 1–4 were different to those of the baseline (*p* < 0.05), while in the fremanezumab experiments all the values were significantly lower than the baseline (*p* < 0.0001). The mean flow during the 5 min of stimulation was 82.4% of the baseline in the control antibody experiments and 63.4% in the fremanezumab experiments, which was significantly different (Student’s *t*-test, df = 20, *p* < 0.05; Figure 6C right). Thus, capsaicin at the high dose of 10 µM caused a transient decrease in flow for 4 min in the rats treated with the isotype control antibody but a robust and sustained decrease in flow in the rats treated with fremanezumab.

## 4. Discussion

The present study was initiated in an attempt to further examine changes in the nociceptor function in the trigeminovascular system induced by the systemic administration of the monoclonal anti-CGRP antibody fremanezumab. We subcutaneously administered the anti-CGRP antibody fremanezumab to rats, in a manner similar to the human clinical use of this antibody for the prevention of chronic and frequent migraine. Rats treated with fremanezumab or an isotype control antibody were studied by applying well-established ex vivo and in vivo experimental models of meningeal nociception relevant to the pathophysiology of migraine headache. Fremanezumab did not cause any changes in behaviour, nor did it interfere with the growth of the animals in comparison with isotype control antibody-treated animals. The isotype control antibody does not bind to CGRP and was used as a negative control to assess the effect of specifically targeting CGRP with fremanezumab.

### 4.1. Sex Difference in CGRP Release

First, we examined the amount of CGRP released from the dura mater in our established ex vivo hemisected rat head preparation. The CGRP release was reduced as early as three days after the fremanezumab treatment, which applied to both the spontaneous (basal) release and the stimulated release following the application of the TRPV1 agonist capsaicin (500 nM), and this effect lasted up to four weeks. Being aware of the possible sex differences in CGRP signalling, we used male and female rats in equal parts. Although we did not test the oestrus cycle of the female animals, the females showed significantly higher basal CGRP release from the dura mater compared to the males; hence, the reduction observed with the fremanezumab treatment on the basal and stimulated CGRP release was more robust in the females than in the males. To our knowledge, a sex difference in CGRP release has not been published so far; however, it has been reported that female compared to male rodents are more sensitive to CGRP [71]. In the latter study [71], only female animals showed facial mechanical hypersensitivity and pain-like grimace behaviour when CGRP at low doses was directly applied onto the dura mater, as well as mechanical hypersensitivity of the hind paws after intra-plantar injection of CGRP, suggesting a generally higher susceptibility to CGRP in females. Earlier, it was reported that systemically or locally administered CGRP did not excite or sensitize meningeal afferents, but in these experiments only male rats were used [23]. In another study with both male and female mice, intraperitoneal CGRP injection caused grimace behaviour indicating pain, without a significant sex difference but with a trend of higher responsiveness in females [72]. In this case, sumatriptan reduced the response to CGRP only in male animals, while the anti-CGRP antibody ALD405 was effective in both sexes. These discrepancies may be partly due to the different modes of application of CGRP and the doses, as discussed [27], and demonstrate that differentiating sexes is important regarding the examination and therapeutic targeting of the CGRP release and signalling system.

### 4.2. Where Do Anti-CGRP Antibodies Act?

The migraine-preventing effect of CGRP-targeting antibodies is considered mainly as a peripheral effect in the trigeminovascular system as antibody penetration of the blood brain barrier (BBB) into the central nervous system is limited. CGRP is also released from the central terminals of the activated trigeminal afferents within the spinal trigeminal nucleus [36,37] and contributes to synaptic transmission [14,73]. Immunohistochemical labelling of a fusion protein of the CGRP receptor components RAMP1 and CLR suggests CGRP binding in the monkey spinal trigeminal nucleus, in addition to the known peripheral sites of CGRP receptor expression in the trigeminovascular system [74]. However, due to the limited access of IgG antibodies to central sites within the BBB, an effective central effect of CGRP-targeting antibodies appears unlikely. Moreover, despite some controversial clinical observations, no convincing evidence supports the assumption that migraine attacks enhance the permeability of the BBB, allowing the passage of chemical substances from the blood into the brain tissue [75]. Therefore, trigeminal ganglion neurons and satellite cells and the dura mater innervated by the peripheral axons of the trigeminal afferents—structures not protected by the BBB—are the most likely targets of the antibody treatment. While it is presumed that the anti-CGRP antibodies act in the periphery, the precise target of peripheral CGRP signalling to afferent structures is not known. Based on the immunohistochemical findings, it has been hypothesized that CGRP released from C-fibres binds to CGRP receptors located on the sensory axons of the Aδ-fibres, namely within the nodes of Ranvier, thereby sensitizing the membrane channels of the Aδ-fibres [76], but functional evidence for this assumption is lacking. The main problem with this hypothesis is that voltage-dependent conduction channels could possibly be sensitized to increased excitatory currents, as seen in experimental neuropathic conditions [77,78], but it does not explain the CGRP release from the adjacent C-fibres. The hypothesis is at least in line with the finding that fremanezumab inhibits mainly the Aδ fibres in a rat model of the cortical spreading depression-induced activation of second-order neurons in the rat spinal trigeminal nucleus [63]. However, the mechanism of CGRP involvement in cortical spreading depression is not yet clear [79]. Fremanezumab does not inhibit the arterial dilatation induced by cortical spreading depression in rat [80].

### 4.3. Meningeal Blood Flow Induced by the Stimulation of TRP Receptors

Our release experiments were complemented by in vivo meningeal blood flow recordings as functional measurements reflecting the blood flow-increasing effect of CGRP released from nociceptive afferents. Consistently with the CGRP release measurements, both the basal blood flow and the flow induced by the TRPA1 agonist acrolein and the low concentration of the TRPV1 agonist capsaicin (100 nM), which are both known to induce CGRP release, were reduced in the animals treated with fremanezumab compared to the isotype control antibody. Capsaicin at the high dose of 10 µM, which directly activates the TRPV1 receptors of the vascular smooth muscle cells, induced a transient decrease in flow for 4 min in the rats treated with the antibody isotype but a robust and sustained decrease in flow in the rats treated with fremanezumab. Topical application of capsaicin at this high dose has been shown to decrease meningeal blood flow [62], most likely by direct activation of the TRPV1 receptors of the vascular smooth muscle cells that induce calcium inflow and smooth muscle constriction [43]. The CGRP-releasing effect of capsaicin from the meningeal afferents counteracts this effect, which is partly effective in rats treated with the isotype control antibody but not in rats treated with fremanezumab.

### 4.4. Possible Effects of Anti-CGRP Antibodies beyond CGRP Neutralization

Anti-CGRP antibodies are thought to neutralize part of the CGRP molecules released by activated peptidergic afferents, thus lowering the capacity of the CGRP signalling. They most likely exert their effect within the tissues such as the dura mater and the trigeminal ganglion, probably directly at the sites of CGRP release. The mechanisms by which CGRP interacts with other nerve fibres is largely unknown, as discussed above; however, the effects of the gene expression within the trigeminal ganglion may contribute to sensitization. The CGRP receptors are expressed by both neurons and satellite glial cells in the trigeminal ganglion [81,82], and signalling between CGRP-releasing and CGRP-sensitive trigeminal ganglion cells seems possible [83]. CGRP may induce gene expression by ERK signalling [84], which could result, for example, in an increase in neuronal NO synthase and hence the production of NO in the ganglion [85]. The NO may back-signal to the CGRP-producing cells, which then increase their expression of CGRP [86] and the CGRP receptor components in other neurons [51]. In short, these mechanisms could induce a vicious circle, including the production of other sensitizing mediators such as brain-derived neurotrophic factor, which then are delivered by axonal transport to the peripheral nerve fibres and central terminals [87]. Anti-CGRP antibodies, which may act in the trigeminal ganglion, could interrupt this cross-signalling at an early state and prevent the peripheral sensitization dependent on the gene expression.

Alexa Fluor 594-conjugated fremanezumab has recently been found not only in the sensory but also in the autonomic ganglia of rats [21]. Early investigations have already revealed CGRP immunoreactivity in small numbers of neuronal cell bodies but in numerous stained axons in several parasympathetic ganglia [88]. Varicose CGRP-immunoreactive nerve fibres have been described as forming synaptic-like contacts with the somata of the sphenopalatine and ciliary ganglia [89,90], and CGRP receptor components have been localized in rat sphenopalatine ganglion, especially in the satellite glial cells [91]. These morphological findings may form the structural basis for the so-called trigemino-parasympathetic reflex [92], which is postulated to crucially contribute not only to the generation of the cluster headache [93] but also to migraine pain [94], but this has not been sufficiently examined so far. The assumed signalling between the CGRP-secreting nerve fibres and the glial cells in autonomic ganglia is certainly not an acute synaptic process but indicates, rather, an indirect control of neuronal processing, which is most likely based on gene regulation in the glia and/or neuronal cells.

## 5. Conclusions

Taken together, our results suggest that fremanezumab is able to neutralize part of the released CGRP in rat dura mater and possibly also in the trigeminal ganglion. In addition, fremanezumab significantly decreases the CGRP release evoked by noxious stimulation, thereby lowering meningeal blood flow. Fremanezumab’s effect on CGRP release and meningeal blood flow is more pronounced in female than in male animals. Our data may provide a mechanism explaining the functional changes in the trigeminovascular system, leading to reduced pain susceptibility in migraine patients treated with CGRP-targeting monoclonal antibodies.

## Figures and Tables

**Figure 1 cells-11-01768-f001:**
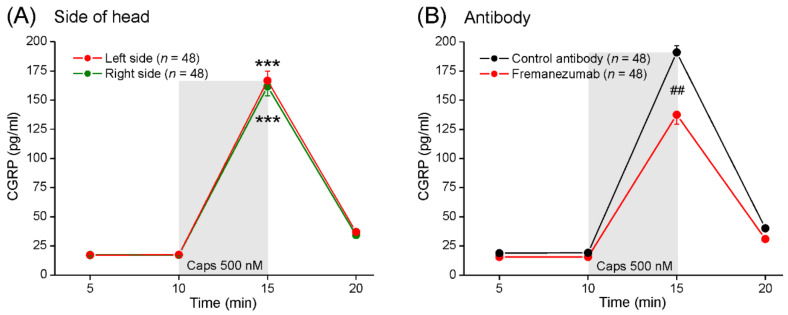
Sequential CGRP release from the dura mater 10 days after injection of antibodies. Data were grouped according to the head side of animals (**A**) and control antibody or fremanezumab treatment, respectively (**B**). Basal release was measured 5 and 10 min after application of physiological solution (SIF) and stimulated release after capsaicin application (Caps 500 nM). Repeated measures ANOVA with Tukey post hoc test, ***: *p* < 0.0001 to basal value (**A**) and ##: *p* < 0.001 between antibodies (**B**).

**Figure 2 cells-11-01768-f002:**
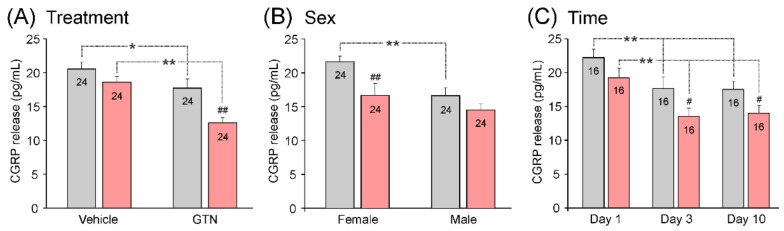
Basal CGRP release from the dura mater of animals treated with control antibody (grey) or fremanezumab (red) depending on injection of vehicle or GTN (**A**), animal’s sex (**B**), and waiting time (day) after antibody treatment (**C**). Data are means ± SEM of averaged release values measured at 5 and 10 min; numbers of experiments shown in the columns; and significant differences between groups (factorial ANOVA and Tukey post hoc test, *: *p* < 0.01, **: *p* < 0.001) as well as between control antibody and fremanezumab (# *p* < 0.05, ## *p* < 0.001).

**Figure 3 cells-11-01768-f003:**
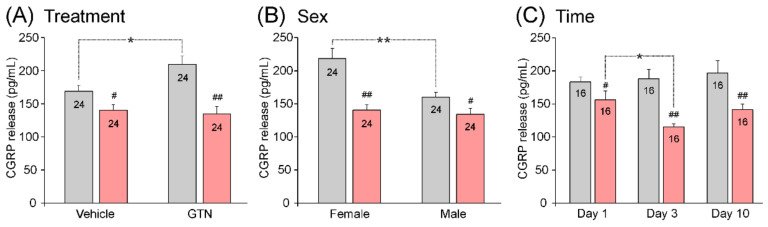
Stimulated CGRP release (mean ± SEM) from the dura mater of animals treated with control antibody (grey) or fremanezumab (red) depending on injection of vehicle or GTN (**A**), animal’s sex (**B**), and waiting time after antibody treatment (**C**). Numbers within columns denote counts of experiments; significant difference between groups (*: *p* < 0.05, **: *p* < 0.001) and between control antibody and fremanezumab (#: *p* < 0.05, ##: *p* < 0.001).

**Figure 4 cells-11-01768-f004:**
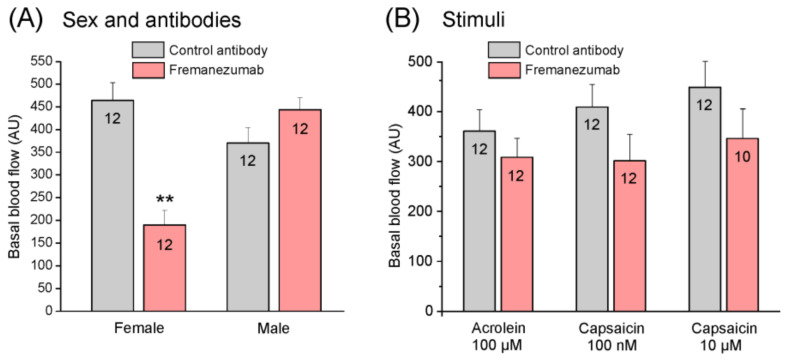
Basal blood flow of the dura mater (means ± SEM) of female and male rats 13–20 days after injection of either control antibody (grey) or fremanezumab (red) (**A**) and before stimulation with acrolein and capsaicin at two doses (**B**). Numbers within columns denote counts of experiments; **: *p* < 0.001 comparing the effect of control antibody and fremanezumab treatment on basal blood flow in females.

**Figure 5 cells-11-01768-f005:**
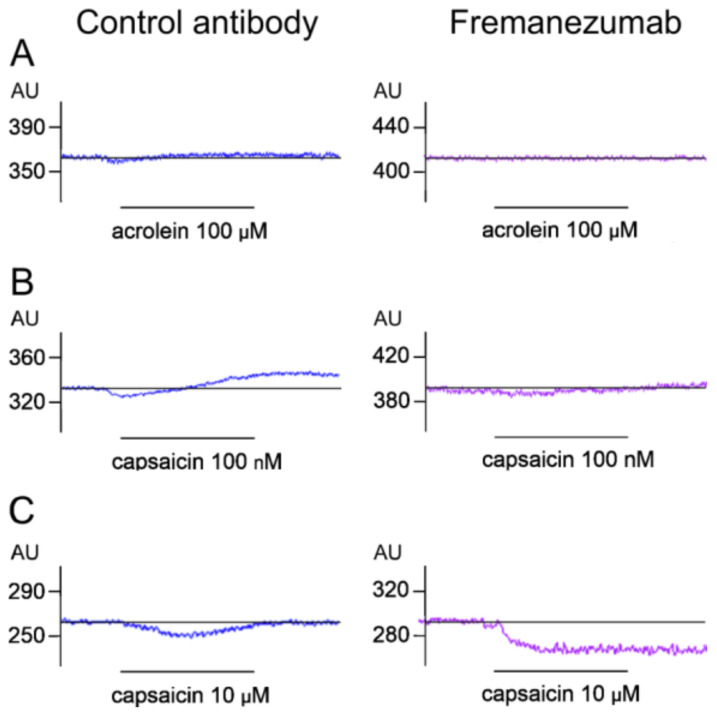
Examples of original blood flow recordings comparing responses to 5 min application of acrolein (**A**), capsaicin at low dose (**B**), and at high dose (**C**) after treatment with control antibody (left) and fremanezumab (right). AU, arbitrary perfusion units.

**Figure 6 cells-11-01768-f006:**
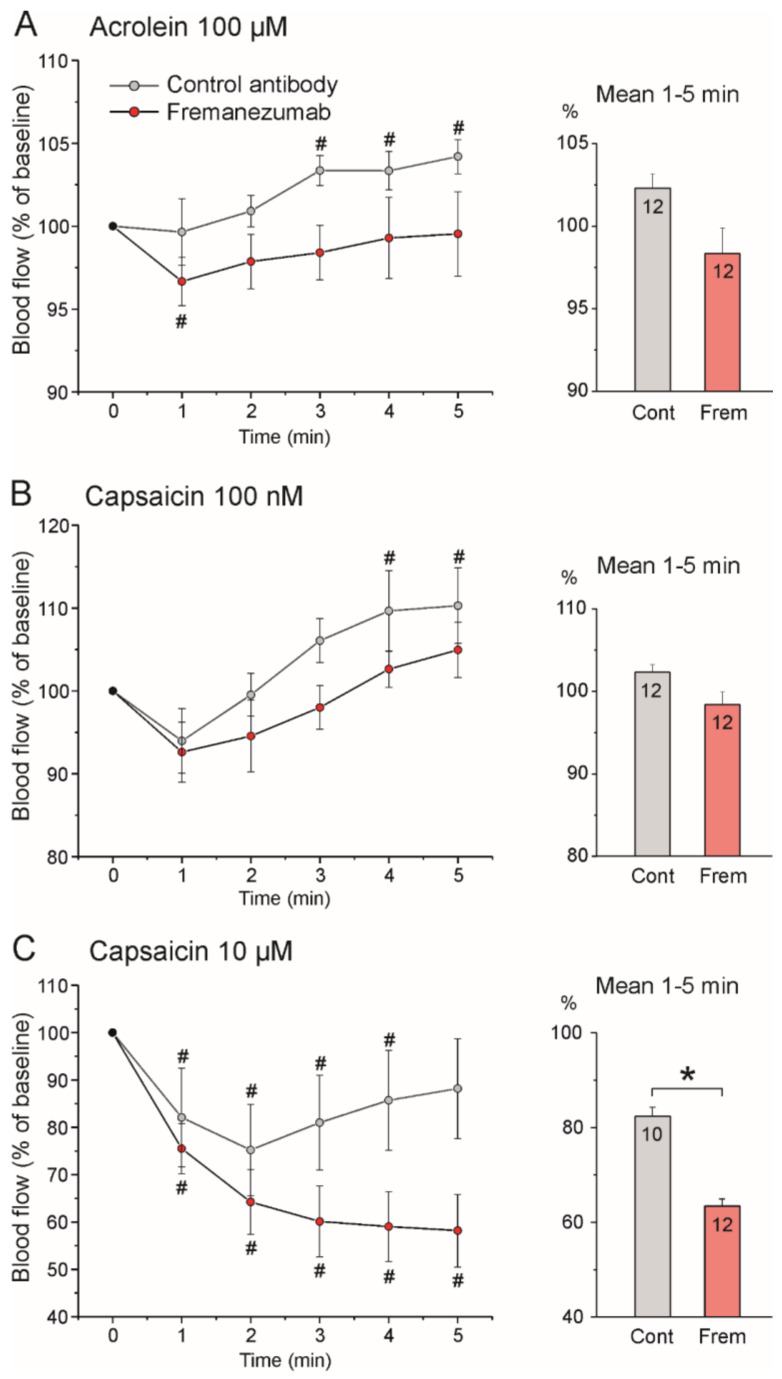
Stimulated blood flow of the dura mater (means ± SEM) of rats of both sexes 13–20 days after injection of either control antibody (grey) or fremanezumab (red) before stimulation with acrolein (**A**) and capsaicin at two doses (**B**,**C**). Blood flow was normalized to the baseline flow prior to stimulation; # significant difference to baseline; * significant difference between control antibody and fremanezumab experiments; numbers within columns denote counts of experiments.

## Data Availability

Not applicable.

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
