# Peer review of "The Anti-CGRP Antibody Fremanezumab Lowers CGRP Release from Rat Dura Mater and Meningeal Blood Flow"

_cells, 2022, doi:10.3390/cells11111768_

Round 1
Reviewer 1 Report
The anti-CGRP antibody fremanezumab lowers CGRP release from rat dura mater and meningeal blood flow by Dux and colleagues.
This is an experimental study in male and female rats to investigate the effect of the anti-CGRP mAb fremanezumab, on capsaicin-induced CGRP release from dura mater and capsaicin- or acrolein-induced meningeal blood flow, after pretreatment with CNT or not. Vehicle rats were used, treated with control Ab. Treatment with fremanezumab 30mg sc attenuated basal and capsaicin evoked CGRP release (days 3-30 postreatment) vs. vehicle treatment. Co-administration of GNT further enhanced the difference. The effect of fremanezumab varied by animal sex, being more robust in female rats. Similarly, treatment with fremanezumab 30mg sc attenuated both capsaicin and acrolein induced meningeal blood flow for 13-20 days post treatment, vs. vehicle animals. Thus, fremanezumab depresses CGRP release for up to four weeks, resulting in CGRP-dependent meningeal blood flow attenuation. These findings indicate that fremanezumab prevents the dural CGRP release induced by noxious cephalic stimulation.
The study is well performed and the article well written. I have a few important comments, however.
- Whether the effect of fremanezumab on capsaicin-induced dural CGRP release varies by animal sex needs more investigation and the data presented here do not clarify the topic, mainly because of the limited number of animals. Since the investigators raised the issue, they must answer it with more experiments, by assessing the oestrus state in addition. Otherwise, they may omit the relevant data.
- Further investigation needs the negative correlation of capsaicin induced dural CGRP and SP release, as well.
- The high and low dose of capsaicin should be investigated in specific pre-experiments to present a curve with different capsaicin doses (0.1, 1, 10, 100, μM) along with the effect on CGRP release and blood flow.
Minor comment: Please report what was the animal mortality related to the experimental procedure.
Author Response
This is an experimental study in male and female rats to investigate the effect of the anti-CGRP mAb fremanezumab, on capsaicin-induced CGRP release from dura mater and capsaicin- or acrolein-induced meningeal blood flow, after pretreatment with CNT or not. Vehicle rats were used, treated with control Ab. Treatment with fremanezumab 30mg sc attenuated basal and capsaicin evoked CGRP release (days 3-30 postreatment) vs. vehicle treatment. Co-administration of GNT further enhanced the difference. The effect of fremanezumab varied by animal sex, being more robust in female rats. Similarly, treatment with fremanezumab 30mg sc attenuated both capsaicin and acrolein induced meningeal blood flow for 13-20 days post treatment, vs. vehicle animals. Thus, fremanezumab depresses CGRP release for up to four weeks, resulting in CGRP-dependent meningeal blood flow attenuation. These findings indicate that fremanezumab prevents the dural CGRP release induced by noxious cephalic stimulation.
The study is well performed and the article well written. I have a few important comments, however.
- Whether the effect of fremanezumab on capsaicin-induced dural CGRP release varies by animal sex needs more investigation and the data presented here do not clarify the topic, mainly because of the limited number of animals. Since the investigators raised the issue, they must answer it with more experiments, by assessing the oestrus state in addition. Otherwise, they may omit the relevant data.
Response: We understand that from a physiological point of view it would be interesting to see if the CGRP release in female animals depends on the estrus and hence the hormonal state, however, in the experiments with long waiting time (up to 30 days) hormonal changes during the experimental time are unavoidable, since the oestrus cycle in rat is around 4 days. The influence of hormonal changes on the CGRP release is certainly very limited, shown by the similar variability (SD) relative to the absolute values in CGRP release in females vs males (SEM basal release: 0.31 vs 0.35; evoked release: 0.35 vs 0.29). We also are not quite sure which different result (more or less difference to males in CGRP release?) the reviewer expects when using specified groups of females according to the 4 phases of the oestrus cycle (https://doi.org/10.1186/s40738-020-00074-3), which would blow up the number of animals. On a clinical point of view, separating the females by their estrus would not be helpful, since anti-CGRP antibodies for the treatment of migraine are given independently of the women‘s hormonal state.
- Further investigation needs the negative correlation of capsaicin induced dural CGRP and SP release, as well.
Response: We agree that the results with CGRP-SP-correlation are too preliminary. Therefore, we have removed this paragraph together with the respective paragraphs in the Methods and the Discussions.
- The high and low dose of capsaicin should be investigated in specific pre-experiments to present a curve with different capsaicin doses (0.1, 1, 10, 100, μM) along with the effect on CGRP release and blood flow.
Response: In previous experiments we have investigated the impact of different capsaicin concentrations on the meningeal blood flow (Dux et al. 2003; ref. 62) and the CGRP release from rat dura mater (Gupta et al. 2010, ref. 70). We have tried to optimize the capsaicin concentration in the present study according to our previous results and chose 100 nM and 10 µM for the flow experiments, since with these concentrations the characteristic increase or decrease, respectively, of the evoked flow was expected. For the release experiments we chose the higher capsaicin concentration of 500 nM to have a sufficient range of lowering CGRP release upon inhibition.
Minor comment: Please report what was the animal mortality related to the experimental procedure.
Response: None of the animals died during the waiting time following injection of antibodies. We have added this statement to the results. The applied drugs had no obvious negative effect on the animal’s health.
We like to thank the reviewer for his/her valuable comments and suggestions.
Reviewer 2 Report
The research work by Dux et al investigates under a complex set of experimental conditions the effect of fremanezumab, one of mAbs against CGRP recently introduced in the clinic of chronic migraine, on GCRP and on Substance P release, on meningeal blood flow and concludes that fremanezumab affects either mechanism, i.e. CGRP release and CGRP-dependent meningeal blood flow as well as may also influence the CGRP release stimulated by noxious agents relevant for the generation of migraine pain. Furthermore, the release of Substance P is inversely correlated with the behaviour of CGRP.
Despite the complexity of the experimental settings and the difficulties of reconciling the numerous data yielded in to an organic discussion, this referee has no major criticisms to raise in order to improve the impact of this research work. This latter conclusion is based on the evidence that: 1) the Authors are experts in the field and have contributed to the development of the experimental model used (i.e. dura mater preparation); 2) there are no concerns for the use of laboratory animals and experimental protocol having received all the approvals from the notified organism, both institutional (University of Erlanghen) and administrative (District Government of Middle Franconia (54-2532.1-99 21/12); 3) appropriate control experiments are carried out and the level of blindness for the execution of the research work is declared; 4) the protocol for EIA assessment of peptides (CGRP, Substance P) concentration is described in detail. Similarly, EIA control experiments for Abs interference with the assay are fully described and convincing.
However, there are aspects of the study that should be clarified for better understanding of the reader.
1) Out of 78 total number of animals (39 males and 39 females), in general, there is no indication of how many animals per experimental group have been used under the diverse experimental conditions. Exception to this, is the report of lack of effect of mAb on the gross behaviour and body weight (assumed the whole of animal used, i.e. 78), side and Abs effect on CGRP release under control condition, the latter being reported with n=48. Only occasionally the number is reported in bracket but it is not clearly referred to animals used or measurements made. Without a clear report of the number of animals used in every experimental condition it is difficult to follow the whole report.
2) In a previous study the Authors reported that “After 1 h treatment of the hemisected skulls with different concentrations of the vehicle (SIF) containing ethanol (0.0316–1.055%), the CGRP concentration in the initial SIF solutions (basal release) varied between 4.4 and 25.3 pg/mL, which is mainly due to the inter-assay variation of the EIA kits (see https://doi.org/10.1186/s10194-021-01235-5)”. Accordingly, values reported here in the range given above should be considered basal. However, at variance with the latter, this range of changes are attributed to the experimental condition (see for instance Abs treatment 15-20 pg/ml) and the basal value is around 8 pg/ml.
3) The description of the blood flow data generation is well detailed. However, it would be better to add a figure showing a typical example of baseline, stimulated and Abs effect on blood flow recordings.
4) If the Authors have carried out behavioural tests for pain assessment these should be incorporated in the text, since they state that “In this study only female animals showed facial mechanical hypersensitivity and pain-like grimace behaviour when CGRP at low doses was directly applied onto the dura mater, as well as mechanical hypersensitivity of hind paws after intra-plantar injection of CGRP, suggesting a generally higher susceptibility to CGRP in females.”
5) The conclusion “We therefore speculate that after fremanezumab treatment less CGRP but more substance P for compensation may be packed into the vesicles of trigeminal ganglion neurons and transported to the periphery.” is quite difficult to consider supported by the data generated because, apparently, in this set of experiments 1) CGRP concentration is far higher than the rest of the whole set of precedent release data; 2) a large number of release experiments to set the threshold range of Substance P is not available as for the case of CGRP and this limits the value of the results.
6) The reasoning “….CGRP releasing effect of capsaicin from meningeal afferents is counteracting this effect, which is partly effective in rats treated with the isotype control antibody but not in rats treated with fremanezumab.” is made complicated by the very high dose of caps used. In other words, the Authors should have also used the 500 nM concentration to envisage relationship with the releasing effect reported here, though under different conditions.
7) Finally, there is more than a hypothesis for a central action of mAbs against CGRP in migraine and this evidence should be discussed by the Authors (see doi: 10.1111/head.13695 for discussion).
Author Response
The research work by Dux et al investigates under a complex set of experimental conditions the effect of fremanezumab, one of mAbs against CGRP recently introduced in the clinic of chronic migraine, on GCRP and on Substance P release, on meningeal blood flow and concludes that fremanezumab affects either mechanism, i.e. CGRP release and CGRP-dependent meningeal blood flow as well as may also influence the CGRP release stimulated by noxious agents relevant for the generation of migraine pain. Furthermore, the release of Substance P is inversely correlated with the behaviour of CGRP.
Despite the complexity of the experimental settings and the difficulties of reconciling the numerous data yielded in to an organic discussion, this referee has no major criticisms to raise in order to improve the impact of this research work. This latter conclusion is based on the evidence that: 1) the Authors are experts in the field and have contributed to the development of the experimental model used (i.e. dura mater preparation); 2) there are no concerns for the use of laboratory animals and experimental protocol having received all the approvals from the notified organism, both institutional (University of Erlanghen) and administrative (District Government of Middle Franconia (54-2532.1-99 21/12); 3) appropriate control experiments are carried out and the level of blindness for the execution of the research work is declared; 4) the protocol for EIA assessment of peptides (CGRP, Substance P) concentration is described in detail. Similarly, EIA control experiments for Abs interference with the assay are fully described and convincing.
However, there are aspects of the study that should be clarified for better understanding of the reader.
1) Out of 78 total number of animals (39 males and 39 females), in general, there is no indication of how many animals per experimental group have been used under the diverse experimental conditions. Exception to this, is the report of lack of effect of mAb on the gross behaviour and body weight (assumed the whole of animal used, i.e. 78), side and Abs effect on CGRP release under control condition, the latter being reported with n=48. Only occasionally the number is reported in bracket but it is not clearly referred to animals used or measurements made. Without a clear report of the number of animals used in every experimental condition it is difficult to follow the whole report.
Response: We have added the number of animals used for the different experiments to the respective paragraphs in the Results. For the main part of the release experiments we have used 48 animals, 24 males and 24 females. Half of these groups (12 males and 12 females) were injected with fremanezumab, the other half (12 males and 12 females) with isotype antibody. Half of the animals from each groups (6 males and 6 females) were injected with GTN before the stimulation with capsaicin. From each rat two skull halves were prepared and separately processed. Since there was no difference between the two skull halves of each animal, the number of measurements was thus doubled, as mentioned in the results section. For the release experiments with long waiting time after fremanezumab (28-30 days) we used 6 male and 4 female animals, as stated in the manuscript; all of them were injected with GTN. For the flow experiments we used 10 animals, 5 males and 5 females. In each animal blood flow was measured with two flow probes positioned on different arteries, which according all our previous yields two independent measurements.
2) In a previous study the Authors reported that “After 1 h treatment of the hemisected skulls with different concentrations of the vehicle (SIF) containing ethanol (0.0316–1.055%), the CGRP concentration in the initial SIF solutions (basal release) varied between 4.4 and 25.3 pg/mL, which is mainly due to the inter-assay variation of the EIA kits (see https://doi.org/10.1186/s10194-021-01235-5)”. Accordingly, values reported here in the range given above should be considered basal. However, at variance with the latter, this range of changes are attributed to the experimental condition (see for instance Abs treatment 15-20 pg/ml) and the basal value is around 8 pg/ml.
Response: In some of the previous studies the basal CGRP release was higher than in the present experiment (e.g. https://doi.org/10.1186/s10194-021-01235-5).The reason was that different batches of the ELISA produce different results. Another reason is the different handling of the preparation depending on the person who prepares the hemisected head. Therefore, basal values between different projects can never be compared but only values within same experiments. In the present experiments we used the same batch and the preparation was always made by the same experimenter to minimize the variation. Due to the low inter-assay variation, we preferred showing absolute values, whereas in some previous studies values were normalized to the baseline.
3) The description of the blood flow data generation is well detailed. However, it would be better to add a figure showing a typical example of baseline, stimulated and Abs effect on blood flow recordings.
Response: We have added a figure (new Fig. 5) showing typical examples of original recordings with acrolein, low and high dose capsaicin stimulation.
4) If the Authors have carried out behavioural tests for pain assessment these should be incorporated in the text, since they state that “In this study only female animals showed facial mechanical hypersensitivity and pain-like grimace behaviour when CGRP at low doses was directly applied onto the dura mater, as well as mechanical hypersensitivity of hind paws after intra-plantar injection of CGRP, suggesting a generally higher susceptibility to CGRP in females.”
Response: In the present study we have not performed behavioural tests. The statement „In this study…“ is related to the study cited in the previous sentence. We have changed the wording to: „In the latter study [71] …“.
5) The conclusion “We therefore speculate that after fremanezumab treatment less CGRP but more substance P for compensation may be packed into the vesicles of trigeminal ganglion neurons and transported to the periphery.” is quite difficult to consider supported by the data generated because, apparently, in this set of experiments 1) CGRP concentration is far higher than the rest of the whole set of precedent release data; 2) a large number of release experiments to set the threshold range of Substance P is not available as for the case of CGRP and this limits the value of the results.
Response: We agree that this data is difficult to compare with the results of the main experiments. This data is too preliminary, and therefore we have removed this paragraph together with the respective paragraphs in the Methods and the Discussions and the old Fig. 5.
6) The reasoning “….CGRP releasing effect of capsaicin from meningeal afferents is counteracting this effect, which is partly effective in rats treated with the isotype control antibody but not in rats treated with fremanezumab.” is made complicated by the very high dose of caps used. In other words, the Authors should have also used the 500 nM concentration to envisage relationship with the releasing effect reported here, though under different conditions.
Response: Results from previous experiments (Dux et al 2003, ref. 62) showed that 100 nM capsaicin is optimal for increasing blood flow by CGRP release and its vasodilatory action, whereas 500 nM is already directly stimulating the vascular smooth muscle directly neutralizing the CGRP-dependent vasodilatation. For the release experiments, where the direct vascular effect of capsaicin is irrelevant, we have used the 500 nM dose with idea that this may increase the range of inhibition through fremanezumab.
7) Finally, there is more than a hypothesis for a central action of mAbs against CGRP in migraine and this evidence should be discussed by the Authors (see doi: 10.1111/head.13695 for discussion).
Response: We agree that a central effect cannot be excluded and have removed the addition „if not exclusively“.
We like to thank the reviewer for his/her valuable suggestions and comments.
Reviewer 3 Report
General note
Please have an English speaker versed in scientific manuscripts edit this manuscript.
Introduction
- 2, L. 55-57: please add a reference for coexpression of TRPV1, TRPA1 and CGRP
Methods
P.3, L 113: What does “animals were weighted” mean?
P.3, L 130: DUX et al 2017 reference isn’t in the numbered order
P.3, L135-137: Concentration description in brackets should contain stock solution concentration, then state “diluted to working concentration with SIF”
P3., L. 146: Which peptidase inhibitors were added
P.4, L. 161: From exactly how many animals were samples also tested for substance P?
P.4, L. 162: either μl or μL , the “L” for Liter should be consistently spelled either lower case of with a capital letter
P.5, L.218: This is the only time i.p. is used. Please define abbreviation in methods (P.3, L.125) and then use consistently thereafter
Results:
Please round the data to 1 digit after the decimal point.
Description of statistics used to analyze data belong in Methods.
Every section in the results starts with excessively detailed methods that should be in the Methods section, only.
P.6, Figure 1: “Data are means ± SEM belongs in methods
- 6, L 290: remove paragraph. Sentence in L291 is conclusion of the data.
- 8, 3.3.7. detailed methods do not belong in the Results section
Discussion:
In general the discussion needs to be more focused on the data and not the methods. A better review of the recent literature on anti-CGRP antibody localization in meninges and points of action needs to be done.
P.13, L.464-467: these causalities are not logical. If estrus cycle had an influence on CGRP release in females, wouldn’t that cause an increased data range?
P.13-14, 4.2. Why wasn’t this investigated in this study by measuring BBB permeability? Please refer to Miller et al., 2016 PMID: 27155150 for detailed experiments on the location of anti-CGRP antibody activity in migraine models
P.14, 4.3. why are you discussing the methods you used for CGRP release when the title of your manuscript “The anti-CGRP antibody fremanezumab lowers CGRP release from rat dura mater and meningeal blood flow” suggests an investigation into the efficacy and point of action of fremanezumab? This should be part of the introduction to explain why GTN was chosen for the present study.
p.15, 4.6. Again, please incorporate Miller et al., 2016 PMID: 27155150 and similar studies in your discussion.
Author Response
We thank the reviewer for careful reading the manuscript and his/her valuable comments.
General note
Please have an English speaker versed in scientific manuscripts edit this manuscript.
Response: Both KDM and JS, who have edited the MS, are native English speaking.
Introduction
P.2, L. 55-57: please add a reference for coexpression of TRPV1, TRPA1 and CGRP
Response: Several papers (e.g. refs. 37, 39, 41, 43) have been cited in the MS showing CGRP release upon stimulation with TRPV1 and TRPA1 agonists, clearly indicating coexistence of TRP receptors and CGRP, which has long been known. However, the populations of trigeminal afferents stimulated by capsaicin or acrolein, respectively, are certainly not identical though they may be overlapping. It is therefore not clear why the reviewer asks for coexpression of TRPV1, TRPA1 and CGRP, and we are not aware of a publication showing this.
Methods
P.3, L 113: What does “animals were weighted” mean?
Response: The wording was changed to “weighed”, thank you for indicating this error.
P.3, L 130: DUX et al 2017 reference isn’t in the numbered order
Response: The correct reference is Ebersberger et al. 1999 (ref. 39), which has been added.
P.3, L135-137: Concentration description in brackets should contain stock solution concentration, then state “diluted to working concentration with SIF”
Response: The SIF is not made as a stock solution but as a fresh solution composed of the molar concentrations exactly as it stands.
P3., L. 146: Which peptidase inhibitors were added
Response: The peptidase inhibitors are part of the commercially ordered assay from Bertin Pharma (“EIA buffer”). Unfortunately, it is not specified regarding the ingredients.
P.4, L. 161: From exactly how many animals were samples also tested for substance P?
Response: We have removed this paragraph together with the respective paragraphs in the Methods and the Discussions and the old Fig. 5, because the other reviewers regarded this data as too preliminary.
P.4, L. 162: either μl or μL , the “L” for Liter should be consistently spelled either lower case of with a capital letter
Response: This was part of the removed paragraph.
P.5, L.218: This is the only time i.p. is used. Please define abbreviation in methods (P.3, L.125) and then use consistently thereafter
Response: Done as requested.
Results:
Please round the data to 1 digit after the decimal point.
Response: Data was rounded to 1 digit.
Description of statistics used to analyze data belong in Methods.
Response: We understand that statistics should be described in the Methods as done, however, since there are different statistical procedures adapted to the different results, we find it useful to repeat the specific methods of analysis in the respective Results sections. We also understand that this is not the preferred style of this reviewer but it has not been criticized by the other reviewers and in previous manuscript reviewers have asked to specify the statistics in the Results.
Every section in the results starts with excessively detailed methods that should be in the Methods section, only.
P.6, Figure 1: “Data are means ± SEM belongs in methods
Response: We have removed the term from the legend of Fig. 1.
- 6, L 290: remove paragraph. Sentence in L291 is conclusion of the data.
Response: We have removed this supernumerary paragraph. The content is sufficiently explained in the Discussion.
- 8, 3.3.7. detailed methods do not belong in the Results section
Response: Paragraph 3.3.7 has been removed together with the other substance P data.
Discussion:
In general the discussion needs to be more focused on the data and not the methods. A better review of the recent literature on anti-CGRP antibody localization in meninges and points of action needs to be done.
P.13, L.464-467: these causalities are not logical. If estrus cycle had an influence on CGRP release in females, wouldn’t that cause an increased data range?
Response: The estrus cycle (in rat about 4 days) may indeed have an influence on CGRP release, however, analyzing female rats in different estrus states is beyond the scope of this examination, and the influence seems to be very limited. Generally, the influence of hormonal changes on the CGRP release may be rather limited, shown by the similar variability (SD) relative to the absolute values in CGRP release in females vs males (SEM basal release: 0.31 vs 0.35; evoked release: 0.35 vs 0.29).
P.13-14, 4.2. Why wasn’t this investigated in this study by measuring BBB permeability? Please refer to Miller et al., 2016 PMID: 27155150 for detailed experiments on the location of anti-CGRP antibody activity in migraine models.
Response: The BBB permeability was not subject of this study and is not relevant for the results, which are restricted to the cranial dura mater of the animals. Miller et al (2016) involves mapping CGRP receptor localization in the primate trigeminovascular system. Not surprisingly, central CGRP receptor localization was noted, however, in contrast to the reviewer’s statement, this study does not investigate the “location of anti-CGRP antibody activity in migraine models“. The primates were not dosed with CGRP antibody. Nevertheless, it has been addressed in the Discussion, and therefore we have added the following together with the reference recommended (new ref. 74): “CGRP is also released from central terminals of activated trigeminal afferents within the spinal trigeminal nucleus [36,37] and contributes to synaptic transmission [14,73]. Immunohistochemical labelling of a fusion protein of the CGRP receptor components RAMP1 and CLR suggests CGRP binding in the monkey spinal trigeminal nucleus, in addition to the known peripheral sites of CGRP receptor expression in the trigeminovascular system [74]. However, due to the limited access of IgG antibodies to central sites within the BBB, an effective central effect of CGRP targeting antibodies appears unlikely”.
P.14, 4.3. why are you discussing the methods you used for CGRP release when the title of your manuscript “The anti-CGRP antibody fremanezumab lowers CGRP release from rat dura mater and meningeal blood flow” suggests an investigation into the efficacy and point of action of fremanezumab? This should be part of the introduction to explain why GTN was chosen for the present study.
Response: We have placed this paragraph into the Introduction and adapted the respective reference numbers.
p.15, 4.6. Again, please incorporate Miller et al., 2016 PMID: 27155150 and similar studies in your discussion.
Response: See above. We refrained from adding more studies addressing possible central actions of monoclonal antibodies, since there is not yet functional evidence.